# Screening for Left Ventricular Hypertrophy Using Artificial Intelligence Algorithms Based on 12 Leads of the Electrocardiogram—Applicable in Clinical Practice?—Critical Literature Review with Meta-Analysis

**DOI:** 10.3390/healthcare13040408

**Published:** 2025-02-14

**Authors:** Agata Makowska, Gayathri Ananthakrishnan, Michael Christ, Matthias Dehmer

**Affiliations:** 1Cardiology, Hospital Centre of Biel, 2501 Biel, Switzerland; 2Healthcare Management, Alfred Nobel Business School Switzerland, 8001 Zürich, Switzerland; gayathri.ananthakrishnan@nobeluniv.com; 3Emergency Department, Cantonal Hospital Lucerne, 6000 Lucerne, Switzerland; 4Department of Computer Science, Distance University of Applied Sciences, 3900 Brig, Switzerland; 5Institute of Biomedical Image Analysis, UMIT TIROL—Private University for Health Sciences and Health Technology, 6060 Hall in Tyrol, Austria

**Keywords:** artificial intelligence, left ventricle hypertrophy, electrocardiogram, deep learning, machine learning

## Abstract

**Background/Objectives:** The increasing utilization of artificial intelligence (AI) in the medical field holds the potential to address the global shortage of doctors. However, various challenges, such as usability, privacy, inequality, and misdiagnosis, complicate its application. This literature review focuses on AI’s role in cardiology, specifically its impact on the diagnostic accuracy of AI algorithms analyzing 12-lead electrocardiograms (ECGs) to detect left ventricular hypertrophy (LVH). **Methods:** Following PRISMA 2020 guidelines, we conducted a comprehensive search of PubMed, CENTRAL, Google Scholar, Web of Science, and Cochrane Library. Eligible studies included randomized controlled trials (RCTs), observational studies, and case–control studies across various settings. This review is registered in the PROSPERO database (registration number 531468). **Results:** Seven significant studies were selected and included in our review. Meta-analysis was performed using RevMan. Co-CNN (with incorporated demographic data and clinical variables) demonstrated the highest weighted average sensitivity at 0.84. 2D-CNN models (with demographic features) showed a balanced performance with good sensitivity (0.62) and high specificity (0.82); Co-CNN models excelled in sensitivity (0.84) but had lower specificity (0.71). Traditional ECG criteria (SLV and CV) maintained high specificities but low sensitivities. Scatter plots revealed trends between demographic factors and performance metrics. **Conclusions:** AI algorithms can rapidly analyze ECG data with high sensitivity. The diagnostic accuracy of AI models is variable but generally comparable to classical criteria. Clinical data and the training population of AI algorithms play a critical role in their efficacy. Future research should focus on collecting diverse ECG data across different populations to improve the generalizability of AI algorithms.

## 1. Introduction

Left ventricular hypertrophy (LVH) is a cardiac condition with an increased left ventricular mass. This is measured in relation to body weight and often results from chronic pressure overload [1]. Early detection of LVH due to arterial hypertension is crucial, as it is an independent prognostic factor for cardiovascular events [2]. Traditionally, electrocardiogram (ECG) criteria, such as the Cornell voltage and Sokolow-Lyon voltage, have been used to diagnose LVH [3]. However, these conventional methods have demonstrated limited sensitivity (approximately 19–25%), potentially leading to missed diagnoses [4].

In recent years, artificial intelligence (AI) and deep learning (DL) techniques have revolutionized medical image analysis and diagnostic processes in cardiology. These advanced computational methods have shown remarkable potential in automatically extracting features from large datasets, significantly improving the performance of various cardiac diagnostic tasks [5,6].

Researchers have developed AI algorithms to detect a wide range of cardiovascular conditions using electrocardiogram (ECG) data. For instance, convolutional neural networks (CNNs) have been employed to analyze voltage-time waveforms from ECGs to recognize patterns associated with left ventricular dysfunction or hypertrophic cardiomyopathy [7]. Other studies have utilized deep learning models to predict paroxysmal atrial fibrillation based on ECG in normal sinus rhythm [8,9,10,11].

The application of deep learning to ECG interpretation presents a promising avenue for enhancing the detection of LVH, potentially overcoming the limitations of traditional diagnostic criteria [4]. Several studies have explored the uses of deep learning algorithms, such as CNNs and long short-term memory (LSTM) networks, in analyzing 12-lead ECGs for LVH detection [12,13]. These approaches have demonstrated improved sensitivity and overall diagnostic accuracy compared to conventional ECG criteria.

Recent research by Shimizu et al. analyzed ECG features’ diagnostic performance based on historical (classical) criteria and machine learning models for LVH detection [14]. While conventional parameters like Cornell voltage showed good accuracy, machine learning (ML) models identified non-traditional ECG parameters as essential predictors of LVH. These included mean QT interval, mean QRS duration, and R wave amplitude in lead I. Both supervised and unsupervised ML models can be used in this subfield [15,16]. Interestingly, advanced techniques like the Shapley additive explanation (SHAP) method revealed that V2/V3 S-wave amplitude and I/V5 T-wave amplitude played essential roles in the AI models’ decision-making process [14].

In this review, we focus on surveying supervised learning techniques. That means one needs labeled training data to predict the class labels using a particular method. We briefly present some of those methods: classical ones are, e.g., K-nearest neighbor (KNN) or support vector machine (SVM) classification [15,16]. KNN usually requires the Euclidian distance between data points to predict the class of a new data point, measuring their similarity. KNN is to be determined by minimizing the classification error using performance measures such as F-measure, precision, recall, and so forth [15] (pp. 211–213). Another classical method for supervised learning is artificial (classical) neural networks. They consist of an input layer, several hidden layers, and an output layer where weights must be determined using backpropagation [15]. While backpropagation does not often work in practice, convolutional neural networks (CNN) have been developed [17] (pp. 372–390). They have been used extensively to learn local structures in complex data and, therefore, they have been more efficient than classical NN because the information of parameters can be shared.

Integrating AI with portable ECG devices presents novel opportunities for widespread, accessible LVH screening. AI applications that analyze ECG can now be used in mobile phones and other telemetry devices, including wearable and implantable recording devices. This convergence of AI and portable technology holds great promise for improving the early detection and management of LVH in diverse clinical settings. AI applications in cardiology can be categorized based on their functionality and the specific cardiac conditions they address. In Table 1, a few examples of them are presented [18,19,20,21,22].

Given the rapid evolution of deep learning techniques and their increasing application in cardiology, a comprehensive review of the current literature on AI-based LVH detection is warranted. This review aims to systematically assess the diagnostic accuracy of deep learning models using 12-lead ECGs for LVH detection and evaluate their potential utility in clinical practice.

## 2. Materials and Methods

This review’s reporting was guided by the standards of the Preferred Reporting Items for Systematic Review and Meta-Analysis (PRISMA) Statement [24]. Eligible studies are randomized controlled trials (RCTs), observational studies, and case–control studies in different settings: primary care clinics, hospitals, and community health centers.

The review protocol was registered in the PROSPERO database (registration number: 531468) and is accessible for public scrutiny via the PROSPERO online platform (https://www.crd.york.ac.uk/PROSPERO/display_record.php?RecordID=531468, accessed on 13 February 2025).

A search was conducted using MEDLINE, CENTRAL, Google Scholar, Web of Science, and Cochrane Library databases. The search strategy is presented in Figure 1.

The first and second steps consisted of screening titles and abstracts, followed by screening the cited articles in full text. The search terms included “artificial intelligence”, “electrocardiogram”, “left ventricle hypertrophy”, “echocardiography”, “deep learning”, “sensitivity and specificity”, “accuracy of electrocardiogram analysis”, and “artificial intelligence algorithm”.

Performance measures for search strategies will be evaluated with search recall (search sensitivity) and precision. The criteria for eligible studies were as follows: (1) studies whose main tasks related to the detection or prediction of left ventricular hypertrophy; (2) studies that developed AI models by utilizing either deep learning or conventional machine learning algorithms; (3) studies that developed AI models for smartphones or using data acquired from mobile devices.

Data collection was performed based on pre-defined study PICO criteria:-Population: adult patients (out- and inpatients) with written electrocardiogram-Intervention: analysis of electrocardiogram-Comparator: human interpretation with imaging modalities-Outcome: sensitivity and specificity of AI algorithms

Data were extracted directly from journal articles and other study reports. Reports were made based on STARD (for diagnostic accuracy studies). Two independent reviewers (A.M. and G.A.) screened the search results to identify articles that closely aligned with the objectives and scope of our review. The publication date, the journal’s impact factor, and the authors’ credentials assessed the credibility and relevance of each article and focused on studies that report accuracy metrics were examined (e.g., sensitivity, specificity, along with 95% confidence intervals (CIs), positive predictive value, negative predictive value).

The selection of studies was performed based on the criteria mentioned below.

Inclusion criteria:

-Studies that utilize ML or DL algorithms with ECG data for diagnosing LVH-Research conducted in the last few years-Studies where LVH diagnosis is confirmed through echocardiography or cardiac MRI.-Research that directly compares AI results with the imaging-based LVH diagnosis.

Exclusion criteria:

-Studies that do not involve ML or DL techniques-Research performed before 2000-Studies lacking standardized datasets for evaluation-Models with limited interpretability-Inconsistency in performance when using new datasets-Concerns related to security, privacy of health data, and collaboration with physicians-Studies using other ECG configurations (e.g., single-lead or 3-lead)-Research unrelated to LVH-Studies with inadequate ECG data quality-Studies using other imaging modalities not specified (e.g., thoracic computed tomography)-Research without a direct comparison between AI and imaging-based LVH diagnosis-Investigations with inadequate sample sizes or methodological flaws.

We conducted a literature review of diagnostic accuracy using Cochrane recommendations and RevMan (review manager); this process involves several key steps:-Definition of title and objective of the review-Searching the diagnostic studies-Selection of included studies and data extraction-Assessment of study quality-Statistical analysis-Interpretation of results and development of recommendations

## 3. Results

Fifteen potentially relevant articles were initially selected for full-length review. Eight articles were removed due to exclusion criteria (Table 2). To mitigate potential bias in LVH quantification, particularly the risk of LVH underestimation associated with cardiac magnetic resonance imaging, we ultimately opted to restrict our analysis to studies employing echocardiography as the primary imaging modality. This methodological decision was made to ensure consistency in LVH assessment and enhance results’ comparability across included studies.

Study characteristics are presented in Table 3. The table compares the results of different studies on LVH, focusing on sample sizes, methodologies used AI models), and predictive performance metrics. Table 4 provides more detailed information about the AI methodologies used in each study, including specific architectural details, optimization techniques, and strategies for addressing common challenges in ECG-based LVH detection.

Study characteristics are presented in Table 3. The table compares the results of different studies on LVH, focusing on sample sizes, methodologies that used AI models, and predictive performance metrics. Table 4 provides more detailed information about the AI methodologies used in each study, including specific architectural details, optimization techniques, and strategies for addressing common challenges in ECG-based LVH detection. Diagnostic accuracy is presented separately in Figure 2 which shows that ENN models show generally high accuracy, whereas CNN models show variable performance; DNN models demonstrate good and consistent performance. SL criteria show consistent but generally slightly lower accuracy compared to AI models.

Figure 3a,b present an assessment of the risk of bias and applicability concerns across several studies. Most studies demonstrate a low risk of bias and low applicability concerns, as evidenced by the prevalence of green plus signs. Salazar (2021) [38] exhibits an unclear rating in the “Flow and timing” category under “Risk of Bias”. Zhao (2022) [12] displays unclear ratings in “Patient selection” under both “Risk of Bias” and “Applicability Concerns”.

Meta-analysis was performed with the RevMan (8.16.0.) program after importing study data (mean, standard deviation, sample size, sensitivity, specificity), presented in Figure 4 and Figure 5. Co-CNN (with incorporated demographic data and clinical variables) demonstrated the highest weighted average sensitivity at 0.84. 2D-CNN models (with demographic features) showed a balanced performance with good sensitivity (0.62) and high specificity (0.82); Co-CNN models excelled in sensitivity (0.84) but had lower specificity (0.71). Traditional ECG criteria (SLV and CV) maintained high specificities but low sensitivities. The ROC curves demonstrate a clear trade-off between sensitivity and specificity across all models, with most curves positioned above the diagonal reference line, indicating better-than-random performance. The curves suggest that Co-CNN architecture achieves the most favorable balance between sensitivity and specificity for LVH detection.

## 4. Discussion

A study by Kwon et al. (2019) was one of the first to utilize a large-scale ECG dataset (21,286 subjects) with artificial intelligence for LVH detection [13]. Their approach incorporated both ECG and demographic information. It set a benchmark for AI-based LVH detection but also revealed the challenges in balancing sensitivity and specificity. It emphasized the importance of considering demographic factors alongside ECG data. The study by Kokubo et al. (2020) highlighted the potential superiority of AI models over traditional criteria while also showing that different AI algorithms can yield varying results [33]. This underscores the importance of model selection in AI-based ECG analysis. A study from Cai et al. (2024) represents a significant advancement in AI-based LVH detection, achieving high accuracy while maintaining a balance between sensitivity and specificity [36]. It reinforces the value of integrating demographic data with ECG analysis. Ryu and colleagues (2023) highlighted the gender disparities in AI-based LVH detection, suggesting that models may need to be tailored or adjusted based on gender for optimal performance [37]. It also introduced a new model architecture that shows promise for ECG analysis. While a study from Zhao et al. (2022) did not achieve the highest overall accuracy, it showed the potential of hybrid deep learning architectures in ECG analysis [12]. The balanced sensitivity and specificity suggest that this approach might be beneficial for initial screening purposes.

These detailed analyses of individual studies reveal the progression of AI-based LVH detection over time, from early large-scale implementations to more sophisticated models incorporating demographic data and novel architecture. They also highlight recurring themes such as the trade-offs between sensitivity and specificity, the importance of demographic factors, and the emerging consideration of gender differences in model performance. This evolution in approach and performance underscores the rapid advancement in the field and the potential for further improvements in AI-based ECG analysis for LVH detection.

### 4.1. Performance Variability and Model Selection

The results indicate that AI algorithms, particularly ENN and some CNN variants, can potentially improve the diagnostic accuracy of LVH detection compared to traditional ECG criteria. However, the variations in performance metrics across the reviewed studies highlight the importance of tailoring model selection to specific clinical questions. Our analysis reveals a clear correlation between demographic factors and performance metrics, underscoring the crucial role of clinical data in enhancing the efficacy of neural network models for LVH detection.

### 4.2. Impact of Clinical Data Integration

Incorporating factors such as age, hypertension history, and other clinical characteristics significantly enhance the sensitivity of AI algorithms. However, reliance on clinical factors alongside ECG data raises essential considerations. Firstly, fairness concerns emerge when using ECGs as the sole diagnostic tool, as noted in studies like Zhao et al. [12]. Incorporating extensive clinical and laboratory variables moves beyond ECG data alone, potentially limiting its utility as a standalone screening tool. Many clinical and laboratory data inputs are often unavailable in routine clinical settings [12], highlighting the need for additional data sources in AI-based diagnostic systems.

These challenges support the hypothesis that hybrid models, integrating multiple data types, may offer more robust analytical capabilities, especially when handling complex datasets with diverse patient characteristics. The studies in this meta-analysis employ various AI techniques for LVH detection, each with distinct strengths and limitations. To provide a comprehensive overview of the AI models utilized in this meta-analysis, the advantages and disadvantages in the context of our topic are presented in Table 5 [4,15,16].

This comparative analysis highlights the diverse approaches in AI-based LVH detection, each offering unique benefits and challenges. The choice of models depends on specific clinical needs, available data, and the desired balance between performance and interpretability. CNNs demonstrate high specificity but may require extensive training datasets. DNNs offer versatility but can be prone to overfitting. ENNs may achieve higher accuracy but with increased computational complexity [4,15,16]. Co-CNNs show high sensitivity but may face limitations in clinical settings where additional patient data are not readily available. Understanding these nuances is crucial for interpreting study results and guiding future research in AI-based LVH detection.

### 4.3. Complexity of LVH Diagnosis

The underlying causes of LVH must be considered, as these can substantially influence diagnostic interpretation. Unexplained LVH, as investigated by Sammani [39], could be linked to conditions such as amyloidosis, hypertrophic cardiomyopathy (a genetic form of LVH), or storage cardiomyopathies. These conditions may exhibit distinct ECG features, including low voltage, which can complicate the diagnosis. Additionally, various clinical conditions associated with LVH may alter electrical properties in the heart, resulting in diverse QRS patterns, such as left axis deviation, fascicular blocks, bundle branch blocks, Q waves, and fragmented QRS [40]. These complexities further highlight the challenge of developing a universal AI model that accounts for the full spectrum of LVH causes.

### 4.4. Imaging Modality Considerations

Another critical factor when evaluating AI algorithms for LVH detection is the choice of imaging modality. Cardiac magnetic resonance (CMR) and echocardiography are the primary imaging techniques offering distinct advantages and limitations. While CMR is considered the gold standard for LVH diagnosis, it may underestimate LVH (measured as left ventricular mass) [41]. Conversely, Echocardiography is more widely available but can be limited by image resolution and operator dependence. Furthermore, CMR and echocardiography have different normative values, complicating direct comparisons between AI models trained on data derived from these modalities.

### 4.5. Challenges in Machine Learning Application

A significant theoretical challenge in applying ML algorithms for LVH detection is input data variability, particularly using different training cohorts. This variability can introduce biases and exacerbate the “black box” nature of ML models, raising concerns about the models’ reliability and generalizability. It is a well-known fact that ML models heavily depend on the quality and representativeness of their training data. Suppose training cohorts are not diverse or balanced. In that case, the models may inherit biases—such as demographic imbalances (e.g., age, sex, ethnicity) or clinical differences (e.g., comorbidities or underlying LVH mechanisms like storage diseases or athlete’s heart). Studies have shown that ML models trained on specific populations perform poorly when applied to external datasets with differing characteristics [42,43,44].

A ‘black box’ in artificial intelligence refers to an AI system whose internal working or decision-making processes are opaque and not easily understandable to humans. Also, a mathematical function or equation of the input and output can rarely be inferred. ‘Black box’ models are typically created directly from large datasets through machine learning algorithms rather than being explicitly programmed. This lack of transparency makes it difficult to understand how the model arrives at its conclusions or identify potential prediction biases. For example, a deep learning model trained on echocardiograms from a specific institution may show variations in performance when applied to datasets from other centers [44,45]. The “black box” nature of many AI models presents clinical interpretation and adoption challenges, prompting research into explainable AI techniques. Variability in training data and the choice of imaging modality for LVH confirmation (echocardiography vs. cardiac MRI) introduce additional complexities in model development and validation.

To mitigate the “black box” problem, researchers are developing explainable AI (XAI) methods, such as Shapley Additive Explanations (SHAP), which can highlight the key features influencing predictions, such as specific ECG parameters related to LVH [46,47]. External validation using independent datasets is crucial to assess the robustness of ML models and minimize the risk of overfitting to particular cohorts. Models consistently performing well across diverse datasets are more likely to have clinical utility and demonstrate broader generalizability.

Another limitation of ML is imbalanced datasets, with a lower prevalence of LVH cases compared to non-LVH cases [4]. This imbalance reflects real-world clinical scenarios but can pose challenges for ML models. A few studies explicitly mentioned using data balancing methods. One study used the Synthetic Minority Oversampling Technique (SMOTE) to address class imbalance [48]. Taconne et al. employed random undersampling of the majority class [16]. In the studies that reported using balancing techniques, there was a general trend towards improved sensitivity for LVH detection, often at the cost of slightly reduced specificity [48,49]. The use or absence of data balancing techniques could potentially influence the reported performance metrics, particularly sensitivity and specificity. This variability in methodology may contribute to the heterogeneity observed in our meta-analysis results. Given these observations, we acknowledge that the inconsistent reporting and application of data balancing methods is a limitation of our review.

## 5. Conclusions

Machine learning (ML) algorithms have been proven beneficial and promising for detecting left ventricular hypertrophy (LVH) with higher sensitivity and specificity than traditional methods. However, their reliance on varied training data introduces challenges related to bias and interpretability. Future research in AI-based LVH detection should prioritize three key areas to enhance the reliability and clinical applicability of these models. First, data collection efforts should focus on gathering diverse ECG data from populations with the exact LVH etiology, including a wide range of demographic characteristics and clinical backgrounds. This approach will improve the generalizability of AI algorithms across different patient populations. Second, the emphasis should be on enhancing model interpretability, enabling clinicians to understand the decision-making process of AI systems. This increased transparency will help build trust and facilitate the integration of AI into clinical practice. Finally, rigorous external validation should be conducted to ensure AI models’ reliability and equitable performance across diverse healthcare settings. By addressing these interconnected objectives, researchers can develop AI models for LVH detection that are accurate, trustworthy, and clinically relevant, ultimately improving the diagnosis and management of LVH across diverse patient populations.

## Figures and Tables

**Figure 1 healthcare-13-00408-f001:**
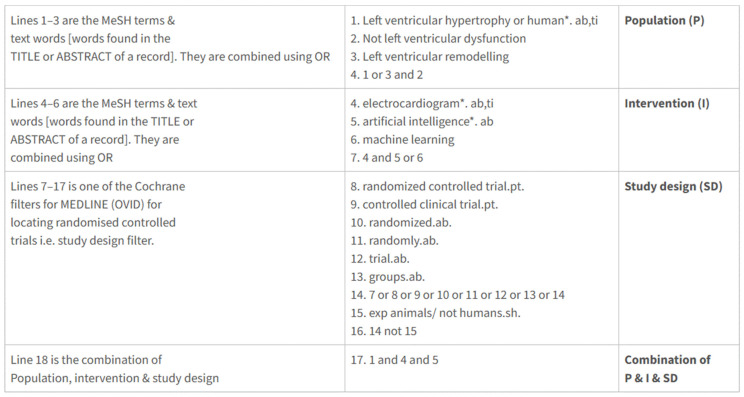
The structure of a search strategy (MEDLINE).

**Figure 2 healthcare-13-00408-f002:**
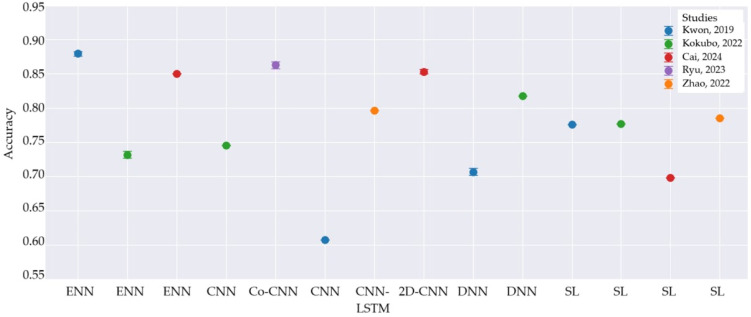
Diagnostic accuracy of AI algorithms and classical ECGs criteria. CNN—Convolutional Neural Network, DNN—Deep Neural Network, ENN—Extreme Learning Machine Neural Network; CNN-LSTM—Convolutional Neural Network—Long Short-Term Memory; SL—Sokolow-Lyon; Kwon, 2019 [34]; Kokubo, 2022 [33]; Cai, 2024 [36]; Ryu, 2023 [37]; Zhao, 2022 [12].

**Figure 3 healthcare-13-00408-f003:**
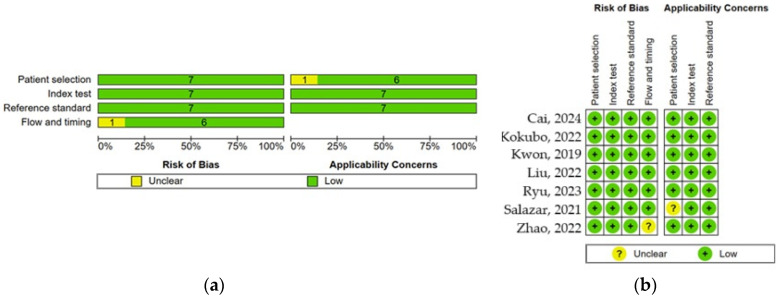
Risk of bias. (**a**) Summary panel; (**b**) Analysis of the studies: Cai, 2024 [36], Kokubo, 2022 [33], Kwon, 2019 [34], Liu, 2022 [35], Ryu, 2023 [37], Salazar, 2021 [38], Zhao, 2022 [12].

**Figure 4 healthcare-13-00408-f004:**
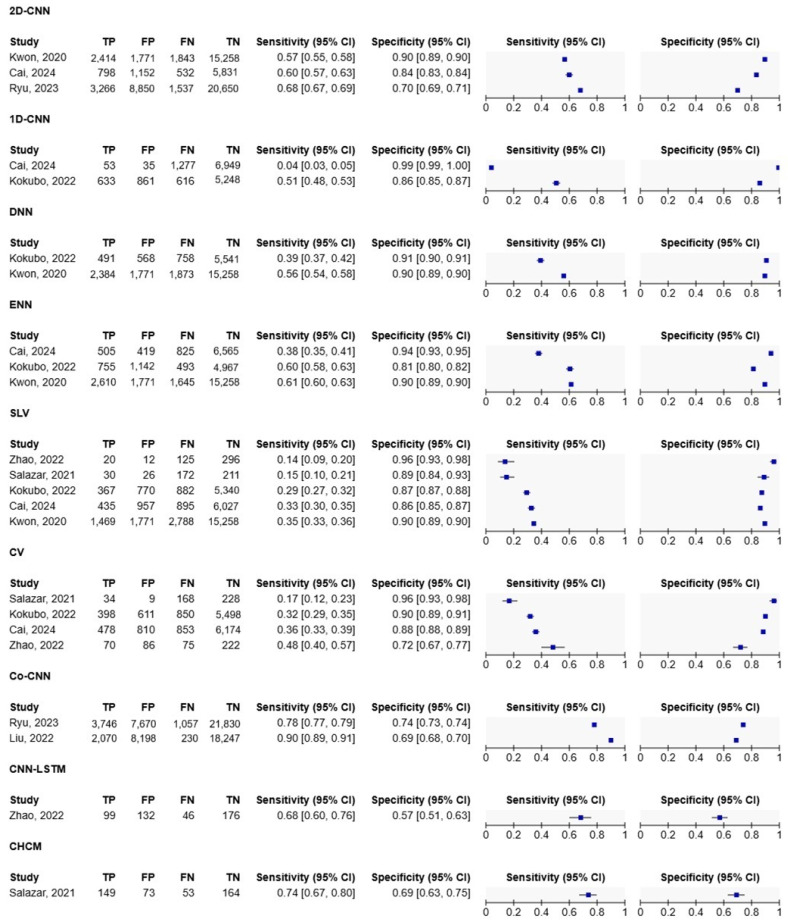
Results of meta-analysis. Cai, 2024 [36]; Kokubo, 2022 [33]; Kwon, 2019 [34]; Liu, 2022 [35]; Ryu, 2023 [37]; Salazar, 2021 [38]; Zhao, 2022 [12].

**Figure 5 healthcare-13-00408-f005:**
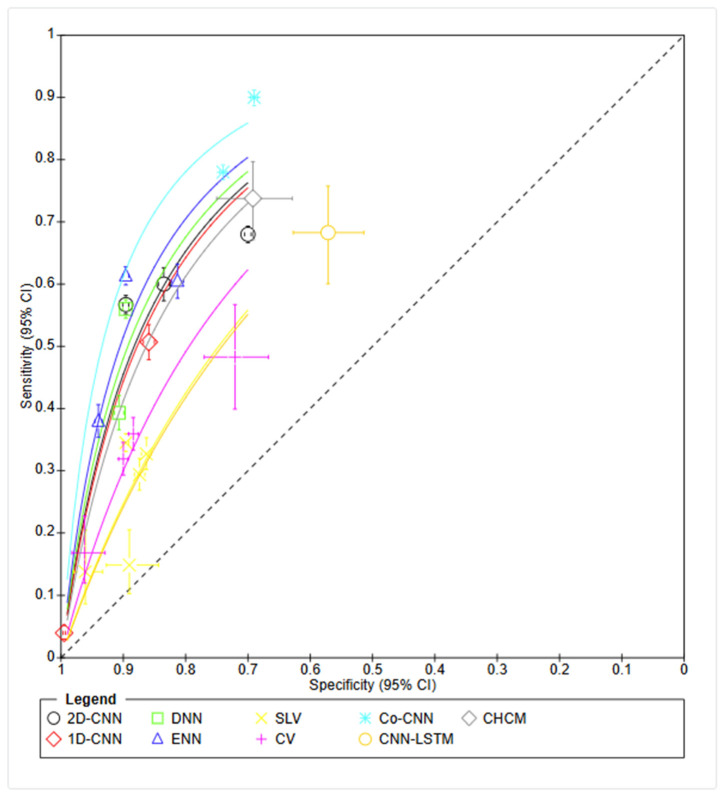
ROC Curve Overview. CNN — Convolutional Neural Network, DNN—Deep Neural Network, ENN—Extreme Learning Machine Neural Network; CNN-LSTM—Convolutional Neural Net-work—Long Short-Term Memory; SL—Sokolow-Lyon.

**Table 1 healthcare-13-00408-t001:** Comparison of available AI applications analyzing ECG.

Application	Advantages	Disadvantages
PMcardio [18]	✓ Validated for diagnosing 39 cardiovascular diseases, including acute coronary occlusion myocardial infarction (OMI) [18,19] ✓ Advanced image recognition for digitizing ECGs (image scan) ✓ personalized treatment recommendations ✓ operates on smartphones ✓ clinical data (i.e., symptoms) are considered in the analyze ✓ 12 canal ECG analyze ✓ CE-marked as a Class IIb medical device (EU MDR 2017/745)	✓ Diagnosis in the form of probability score (low, intermediate, and high) ✓ Potential cost considerations for users (only five examples are free) ✓ Limited to European Union availability ✓ Requires stable internet connection
LabChart ECG Analysis Module [20]	✓ Automatic detection of PQRST waveforms, real-time analysis, supports various animal models ✓ aids in identifying cardiovascular issues like arrhythmias, customizable settings for optimal waveform identification ✓ useful for implantable telemetry devices and intracardiac electrodes ✓ 1 canal ECG analyze ✓ Compliant with GLP and 21 CFR Part 11 regulations	✓ Potential inaccuracies in detecting certain waveforms may require manual adjustments for specific cases, and some traces may be discarded due to detection errors ✓ Specific for arrhythmia, not for other cardiac diagnoses (e.g., LVH or left ventricular dysfunction)
SonoHealth [21]	✓ Portability and convenience ✓ Compact design Standalone functionality without a smartphone ✓ Stores up to 100 readings ✓ FDA approved	✓ Limited diagnostic capability ✓ Concerns about the accuracy of rhythm detection ✓ Not suitable for reliable personal mobile ECG monitoring
KardiaMobile [22]	✓ Offers single-lead and 6-lead options ✓ Integrates with healthcare systems ✓ validated for accurately measuring QT intervals in collaboration with Mayo Clinic [23] ✓ FDA-cleared for detecting various arrhythmias	✓ ECG quality may be lower than standard 12-lead ECGs ✓ Some intervals may be shorter than in standard ECGs ✓ Limited sensitivity for pacemaker rhythms

FDA—The United States Food and Drug Administration.

**Table 2 healthcare-13-00408-t002:** Excluded studies.

Excluded Study	Reason for Exclusion
Haimovich, 2023 [25]	Discrimination of echocardiographic LVH and multiple cardiac diseases associated with LVH. LVH detection by ECG was not examined.
Pan, 2024 [26]	CT-derived LVM, not echocardiographic estimation
Khurshid, 2021 [27]	The deep learning algorithm estimates CMR-derived LV mass with fair accuracy using 12-lead ECG
Naderi, 2023 [28]	Case-to-case comparisons with echocardiography and ECG-AI analysis are missed
Pandelitis, 2023 [29]	Only proof-of-concept study
Shimizu, 2023 [30]	Only the random forest method was used. Sensitivity and specificity not known
Soto, 2022 [31]	Multimodal training models (the electrocardiogram and echocardiogram data were paired according to unique patient identifiers)
Wu, 2020 [32]	The use of six-layer deep neural networks, the electrocardiographic left ventricular hypertrophy classifier (ELVHC), has not been confirmed directly with echocardiography or cardiac MRI.

**Table 3 healthcare-13-00408-t003:** Included studies and their characteristics.

Study	Sample Size	Training Sample	Validation Sample	LVMI (g/m^2^)	LVH (Number of Patients)	Prevalence (LVH Sample/Sample Size)	Used AI Algorithm/ECG Criteria	Predictive Performance Variables
Kokubo, 2022 [33]	7358	3881	943	92.66 ± 96.91 (all patients in the test sample)	1249	0.17	CNN, DNN, ENN, Logistic Regression, Random Forest/ Sokolow-Lyon voltage, Cornell voltage	Sensitivity Specificity Accuracy PPV NPV
Kwon, 2019 [34]	21,286	35,694	3162 (internal) 5476 (external)	67.61 ± 12.46	4353	0.2	CNN, DNN, ENN/ Sokolow-Lyon voltage, included clinical variables	Sensitivity Specificity Accuracy PPV NPV
Liu, 2022 [35]	5749	23,996	3280 (internal) 225 (external)	-	2435	0.08	CNN and a multimodality module that combines the demographic features	Sensitivity Specificity
Cai, 2024 [36]	8314	28,855	3370	111.96 ± 104.09	1363	0.16	CNN, ENN/ Sokolow-Lyon voltage Cornell voltage	Accuracy Sensitivity Specificity
Ryu, 2023 [37]	34,302	24,008	-	146.86 ± 12.23 (men) 126.96 ± 15.54 (women)	4873	0.14	Model with the Entire dataset (with demographic features)	Sensitivity Specificity PPV NPV
Salazar, 2021 [38]	132	307	156	134.2 ± 29	203	0.46	Cardiac Hypertrophy Computer-based model/ Sokolow- Lyon voltage, Cornell voltage	Sensitivity Specificity Accuracy PPV NPV
Zhao, 2022 [12]	453	1120	371	129.28 ± 28.93	144	0.32	CNN-LSTM/ Sokolow-Lyon voltage, Cornell voltage, with demographic features	Accuracy Sensitivity Specificity

LVMI—left ventricular mass index; LVH—left ventricular hypertrophy; AI—artificial intelligence; ECG—electrocardiogram; CNN—Convolutional Neural Network, DNN—Deep Neural Network, ENN—Extreme Learning Machine Neural Network; PPV—Positive predictive value; NPV—Negative predictive value; CNN-LSTM—Convolutional Neural Network—Long Short-Term Memory.

**Table 4 healthcare-13-00408-t004:** Comparison of AI models utilized for each selected study.

Study	AI Model	Input	Structure	Optimization Techniques/Parameters
Kokubo, 2022 [33]	CNN	raw ECG data (1-ECG lead)	- 6 temporal layers; each consist of: CL, BNL, ReLU and MPL - last temporal layer (spatial convolutional layer) is a fusion layer (from 1- to 12-ECG leads) - 2 fully connected layers consist of: BNL, ReLU, DL - number of channels: 16, 16, 32	✓ optimizer: Adam (learning rate, 0.00005) ✓ loss function ✓ batch size: 128 ✓ kernel size: 5, 5, 5, 3, 3
	DNN	19 parameters comprising ECG features and demographic information	- 4 fully connected layers: BNL, ReLU, DL - number of channels: 512, 256, 128, 64	
	ENN	CNN + DNN are combined directly after the convolution layer	- 2 fully connected layers: BNL, ReLU, DL - activation function: sigmoidal (BCEwithLgitsLoss) - number of channels: 16, 16, 32, 32, 64, 64	
Kwon, 2019 [34]	CNN	raw ECG data (12-ECG lead, recording 10 s); 12 × 5000 numbers; 500 Hz - demographic data (sex, age, weight, height) - preprocessing of ECG raw data	- 9 temporal layers: 3 × 2 CL, 2× MPL - The ninth stage is a one flattened layer that converted 3-dimensional feature map information to a 1-dimensional layer of nodes - number of channels: 16, 32, 128, 256, 512	✓ optimizer: Adam ✓ backpropagation method ✓ batch size: 128 ✓ used building library: TensorFlow (the Google Brain Team, Mountain View, CA, USA)
	DNN	11 variables (age, sex, weight, height, heart rate, presence of atrial fibrillation or atrial flutter, QT interval, QRS duration, QTc, R axis, and T axis)	- 5 fully connected layers (i.e., ReLU) - activation function: sigmoidal - number of channels: 12, 12, 12, 12, 8	
	ENN	The last hidden layer of the DNN and the last layer after the flattened layer of CNN	- CNN and DNN are connected to the 1-dimensional layer comprising 520 nodes - number of channels: 12, 8	
Liu, 2022 [35]	CNN with multimodality module	full-length ECG record divided into separate heartbeats (one feature extraction layer; recording 10 s; 500 Hz for 12-ECG leads	- 16 temporal layers: - 8 residual blocks with max-pooling and a skip connection, consist of: BNL, ReLU, EL, DL, DFCL - 1-dimensional convolutional layer that convolves along the time axis using the 12 leads as variates.	✓ kernel size: 25 = 0.05 s of the ECG signal ✓ batch size: 128 ✓ Ten-fold cross-validation (reduces the influence of the LVH prevalence) ✓ Butterworth high-pass and low-pass filter (reduce artifacts) using Biosppy 0.6.1. ✓ ResNet-based model
Cai, 2024 [36]	2D-CNN	1-lead ECG data 240 × 240 pixels), recording 10 s; rate of 500 Hz; 12-lead ECG data converted to 2D images (224 × 224 pixels)	- 12 temporal layers, consti of: CL, BNL, ReLU, MPL - 12 flattened layers - fully connected layers - SoftMax output layer	✓ like GG-Very-Deep-16 CNN(VGG16) ✓ ResNet-based model ✓ 5-fold validation ✓ left, center, and right superposition methods for detecting LVH
Ryu, 2023 [37]	CoAt-Mixer (CNN variant)	12-lead ECG, duration of 10 s, 500 Hz, 12 × 5000 numbers; demographic features	- ‘vanilla structure’: convolutional and transformer elements - Self-attention mechanism for capturing long-range dependencies - pooling and flattened layers - Patch Embedding - GeLU (activation function) - gradient-weighted class activation (Grad-CAM) - Output with 6 class of LVH probability - number of channels: 800, 100, 25	✓ mixup augmentation for improved generalization ✓ employed Shapley Additive Explanations (SHAP) for model interpretation ✓ designed based on CoAtNet and the Conv-mixer ✓ batch size: 768 (feature map A) ✓ work together with ResNet-CBAM ✓ distinguish to female and men
Salazar, 2021 [38]	Computer-based ECG model	458 ECG standard and non-standard parameters; 25 mm/s velocity and 10 mm/mV sensitivity	- ECG computer-based data (Philips DXL-16 algorithm) and the C5.0 ML algorithm - based on decision tree structure: - first step: T voltage I (cut-off: 0.055 mV) - second step: QRS PPK aVL or aVF (cut-off: 1.235 mV and 0.178 mV, respectively) - gradient boosting machine (XGBoost) as the core algorithm - Hyperparameter tuning via grid search	✓ feature selection using LASSO regression ✓ focused on optimizing traditional ECG criteria ✓ 5-fold cross-validation ✓ employed SHAP values for feature importance analysis
Zhao, 2022 [12]	CNN-LSTM hybrid	sliding window technique for ECG signal processing (window size: 250 samples), 1000 Hz, duration of 5 s; 12 × 5000 numbers	- 4 CLs for spatial feature extraction - time distributed layers - LSTM: 2 layers with 128 units each for temporal dependencies - number of channels: 16, 24, 32, 48, 56, and 64	✓ focal loss to address class imbalance ✓ data augmentation: Gaussian noise addition and time warping ✓ 5-fold cross-validation ✓ kernel size: 16, 16, 16

TP—training process; CNN—Convolutional Neural Network, DNN—Deep Neural Network, ENN—Extreme Learning Machine Neural Network; CNN-LSTM—Convolutional Neural Network—Long Short-Term Memory; CL—convolution layer; BNL—batch normalization layer; RELU—rectified linear unit activation function layer; MPL—max-pooling layer; DL—dropout layer; EL—embedding layer; DFCL—dense fully connected layer; GeLU—gaussian error linear unit; QRS PPK, aVL: Peak-to-peak QRS complex amplitudein aVL; QRS PPK, aVF: Peak-to-peak QRS complex amplitude in aVF; ML—machine learning; s—second.

**Table 5 healthcare-13-00408-t005:** Comparison of AI models.

AI Model	Advantages	Disadvantages
CNN	✓ Excellent at identifying spatial patterns in ECG waveforms ✓ Can learn hierarchical features directly from raw ECG signals ✓ Reduced need for manual feature engineering	✓ Require large amounts of training data ✓ Limited interpretability (“black box” nature) ✓ May overfit to training data if not adequately regularized
LSTM Networks	✓ Can model long-term dependencies in time series data ✓ Effective at capturing beat-to-beat variations in ECG	✓ More complex to train than simple CNNs ✓ May struggle with very long sequences
Co-CNN	✓ Demonstrated the highest weighted average sensitivity for LVH detection ✓ Excelled in sensitivity but had lower specificity compared to other models ✓ The inclusion of demographic data and clinical variables enhances the model’s performance	✓ Requires additional clinical data, which may not always be available ✓ More complex model architecture compared to standard CNNs ✓ Potential for bias if clinical variables are not representative of diverse populations
ENN	✓ Combines strengths of multiple models (e.g., CNN and DNN) ✓ Often achieves higher accuracy than individual models ✓ More robust to noise and outliers in the data	✓ Increased computational complexity and training time ✓ Can be more challenging to implement and maintain ✓ May require more storage space for multiple model components
DNN	✓ Capable of learning complex, non-linear relationships in ECG data ✓ Can integrate multiple data types ✓ Flexible architecture allows for customization of specific tasks	✓ Prone to overfitting, especially with limited data ✓ “Black box” nature makes it challenging to explain predictions ✓ Requires careful hyperparameter tuning for optimal performance

CNN—Convolutional Neural Networks; LSTM—Long Short-Term Memory; Co-CNN—Convolutional Neural Network with Clinical Variables; ENN—Ensemble Neural Network; DNN—Deep Neural Network.

## Data Availability

The datasets generated and analyzed during the current review are available from the corresponding author upon reasonable request.

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
