# Peer review of "Screening for Left Ventricular Hypertrophy Using Artificial Intelligence Algorithms Based on 12 Leads of the Electrocardiogram—Applicable in Clinical Practice?—Critical Literature Review with Meta-Analysis"

_healthcare, 2025, doi:10.3390/healthcare13040408_

Round 1

Reviewer 1 Report

Comments and Suggestions for Authors

The review paper is very well done, methodologically sound, and addresses an important issue at the intersection of artificial intelligence and cardiology. There is a very small percentage of studies demonstrating the advantages of using neural networks compared to classical machine learning. The results are presented very clearly and concisely, with the only exception being Figure 5, where the captions should be adjusted to avoid overlapping. The paper lacks a discussion on the specific artificial intelligence techniques used in these studies, why these particular methods were chosen, and their advantages and disadvantages. The only topic addressed is the "black-box" issue, which should be discussed more broadly. Additionally, it would be beneficial to know if any data balancing methods were used in the studies, and if so, which ones, and what impact they might have on the results.

Reviewer 2 Report

Comments and Suggestions for Authors

Dear authors your research methodology and analytical approach are commendable, I have several suggestions to enhance the manuscript's overall quality and readability:

The manuscript would benefit from significant restructuring to achieve better flow and coherence:

  1. Abstract Format The current bullet-point style abstract deviates from standard academic journal conventions. I recommend rewriting it in a continuous paragraph format, maintaining the essential IMRAD (Introduction, Methods, Results, and Discussion) structure while ensuring smooth transitions between components.
  2. Content Flow The current organization shows signs of being adapted from presentation slides, resulting in fragmented discussion and redundant information. For example:

·         The introduction of deep learning concepts appears disjointed, with initial mentions around line 76 followed by a seemingly fresh introduction later

·         The discussion of AI applications lacks smooth progression from general to specific applications

Technical Writing

  1. Many sections, particularly lines 128-163, require reformatting to improve readability and maintain academic writing standards. Consider breaking down complex sentences and ensuring logical progression of ideas.
  2. There are numerous sentence case inconsistencies throughout the manuscript, particularly in section headings and technical terms. These should be standardized according to journal guidelines.

Format and Presentation

  1. Reorganize sections to follow a more traditional academic paper structure, ensuring smooth transitions between topics.
  2. While the figures and tables are informative, their integration into the text could be improved to better support the narrative flow.

Comments on the Quality of English Language

The authors fluctuate between formal academic fullspeak, and the occasional markedly less formal envocation, which has an impact on tone throughout and they use too much active voice like "we" , "we", 
Some parts of the paper is like talking with collegues, needs to be written in a proper approach. Like 166 "We searched for the articles in the MEDLINE and CENTRAL databases." , "We registered".

Some sentences are not fit in to the papers overall flow. like "This is a Literature Review using PRISMA 2020 guidelines"

We registered

Reviewer 3 Report

Comments and Suggestions for Authors

- The paper explains about few studies in the literature about LVH detection using ECG.

- Introduction section especially paragraph 2 and 3 needs to be cited properly. There are citation numbering and citation format issues present. Please look into the complete manuscript for any such issues. Here is a list of related articles that could be used to cite:

     - State-of-the-Art Deep Learning Methods on Electrocardiogram Data: Systematic Review
    - Deep learning for ECG Arrhythmia detection and classification: an overview of progress for period 2017–2023

- The Material and Methods section should specify methods that were used in each selected study for classification etc.
- Lines 166-173 are bold. There are formatting issues in this manuscript, which should be resolved.
- Text should not have capitalized letters in between e.g. line 175.
- There is formatting issue in Tables.
- There are formatting and font issues in Figures.
- It is better to include accuracy rates as well for the studies.
- Results and Discussions are very broadly discussed. However, it is better to include in detail discussion on each study itself.
- A review article usually has many more references. It is better to revise from this perspective as well.

Round 2

Reviewer 3 Report

Comments and Suggestions for Authors

- Table 5 should explain about the AI models utilized for each selected study in your manuscript, not an overall comparison of different AI models.

- You should see other articles that mention different parameters as metric for evaluation and improve Table 4 as it is very inconvenient to read and understand for readers.

Comments on the Quality of English Language

It is fine. Not bad.
